# Advancing and Benchmarking Personalized Tool Invocation for LLMs

## Abstract

Tool invocation is a crucial mechanism for extending the capabilities of Large Language Models (LLMs) and has recently garnered significant attention. It enables LLMs to solve complex problems through tool calls while accessing up-to-date world knowledge. However, existing work primarily focuses on the fundamental ability of LLMs to invoke tools for problem-solving, without considering personalized constraints in tool invocation. In this work, we introduce the concept of Personalized Tool Invocation and define two key tasks: Tool Personalization and Parameter Personalization. Tool Personalization addresses user preferences when selecting among functionally similar tools, while Parameter Personalization considers cases where a user query lacks certain tool parameters, requiring the model to infer them from the user profile. To tackle these challenges, we propose **PTool**, a data synthesis framework designed for personalized tool invocation. Additionally, we construct **PTBench**, a benchmark to evaluate personalized tool invocation. We then fine-tune various open-source models, demonstrating the effectiveness of our framework and providing valuable insights. Our model, training data, and the benchmark will be publicly released upon acceptance.

## 1 Introduction

Recently, large language models (LLMs) have demonstrated remarkable capabilities in natural language processing tasks, particularly in human-computer interaction, where they can effectively comprehend user queries and provide reasonable responses (Zhao et al., 2023). However, the knowledge embedded within LLMs is not inherently up-to-date, as updating these models requires extensive retraining with large-scale data, which incurs significant time and economic costs. To equip LLMs with the ability to solve complex problems and access the latest information, tool invocation capabilities are essential. For instance, LLMs can leverage mathematical tools to decompose and solve intricate mathematical problems or utilize internet APIs (Liu et al., 2025; Qin et al., 2024) and search engines (Schick et al., 2024; Nakano et al., 2021) to retrieve the most recent knowledge.

Existing research on enhancing LLMs's tool invocation abilities primarily focuses on improving fundamental capabilities (Qin et al., 2024; Yan et al., 2024; Lin et al., 2024), such as ensuring adherence to the required syntax, comprehending tool functionalities, interpreting explicit user instructions, and extracting tool parameters. However, in real-world applications, user intents are often implicit rather than explicitly stated, requiring models to infer based on personalized profiles. Recent studies have begun to explore personalized tool invocation (Xu et al., 2025; Cheng et al., 2025), yet a systematic definition is still lacking. Existing work typically relies on idealized assumptions about user profiles–assuming that users' implicit preferences are observable to the model. However, such preferences are embedded in behavioral histories and are rarely provided explicitly in practice.

In this work, we first provide a systematic formulation of personalized tool invocation and highlight two common concepts that exemplify this challenge: **(1) Tool Personalization**. When multiple tools offer similar functionalities, users often exhibit specific preferences. For example, in online shopping, users may choose different platforms depending on their preferences for particular product categories. Some users may prioritize platforms with superior maintenance services when purchasing high-value electronic products, despite the higher cost, while preferring platforms with faster delivery when buying inexpensive daily necessities. Inferring such preferences necessitates reasoning from user attributes, such as age, interests, and purchasing behavior. **(2) Parameter Per-**

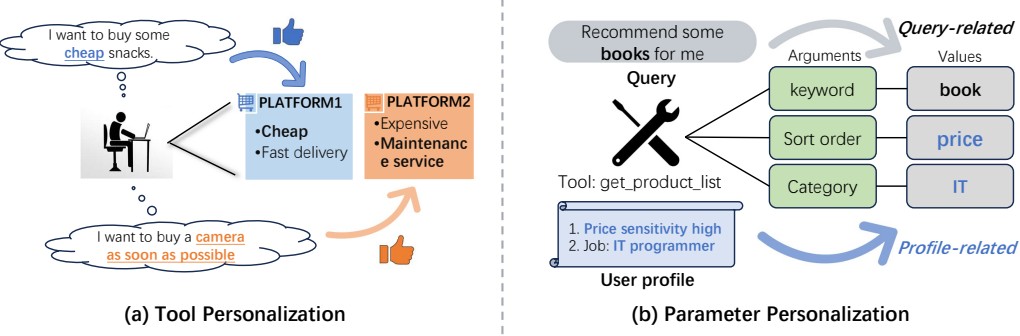

Figure 1: Examples of Personalized Tool Invocation. (a) Tool Personalization: Users may prefer different tools with similar functionalities. (b) Parameter Personalization: Certain tool parameters may be missing from the user's query and must be inferred from the user's profile.

**sonalization**. In everyday scenarios, users tend to express their needs concisely and omit crucial details. For instance, a user might simply request, "Order me a hamburger from KFC", without specifying essential information such as the delivery address, recipient contact details, or preferred delivery time. This requires the model to infer the missing information from the user profile, such as the user's work location, current time, and phone number, ensuring a seamless and accurate tool invocation process. To enhance and systematically evaluate a model's ability in personalized tool invocation, we further introduce an automated data synthesis framework for this task, termed **PTool**, which consists of three key stages: tool generation, user profile construction, and user behavior simulation. Firstly, we consider several commonly used real-world scenarios, where each scenario contains multiple functionally similar platforms organized in a hierarchical tree structure. We then leverage an advanced LLM to recursively decompose platform functionalities using a depth-first expansion approach, progressively refining them until distinct tools are defined for each functional category. Secondly, we abstract and summarize platform features and API parameters to extract both basic user attributes and personalized characteristics, including psychological traits and behavioral tendencies. To construct a diverse set of user profiles, we employ a bottom-up clustering approach for feature induction and a top-down assignment strategy for attribute allocation. Finally, we exploit the role-playing capabilities of LLMs to simulate user behaviors based on the assigned user profiles, generating both historical interactions and potential user queries. To establish reliable ground-truth labels, we further integrate a multi-agent framework that conditions query generation on user profiles. Following manual review and annotation, we construct **P**ersonalized **T**ool**Bench** (**PTBench**), a benchmark designed to evaluate large models' ability in personalized tool invocation, consisting of 1,301 high-quality annotated data samples. Key contributions are summarized as follows:

- We propose a role-playing-based paradigm for personalized tool invocation towards real-world applications, incorporating both user tool personalization and parameter personalization.

- We develop a systematic personalized data synthesis framework and construct PTBench, a benchmark for personalized tool invocation, enabling a comprehensive evaluation of the ability to invoke tools based on user information.

- We demonstrate that training open-source models on our synthesized dataset significantly improves personalized tool invocation capabilities, while also enhancing general tool invocation without compromising other general abilities.

## 2 RELATED WORK

### 2.1 TOOL INVOCATION

Tool invocation (also termed tool calling) involves tool selection from candidate tools and parameter extraction from queries. Existing works can be categorized into two tuning-free and tuning-based methods (Qu et al., 2025; Liu et al.). Tuning-free methods mainly rely on the prompt strategy

with few-shot learning, involving encouraging LLM to reason by providing examples (Yao et al., 2022), rewriting tool documentation with LLMs to enhance the comprehension (Yuan et al., 2024), summarizing tool description with more concise and precise sentence (Xu et al., 2024), leveraging multi-agent collaboration to decompose the tool-calling task (Shi et al., 2024). Tuning-based methods leverage tool-learning samples to train existing LLMs, where the research problems comprise data collection and training strategy. Toolformer (Schick et al., 2024) and ToolkenGPT (Hao et al., 2024) add a special tool-related token into the vocabulary, switching the decoding process into tool selection and calling. Some works leverage advanced LLM to synthesize tool-calling samples to improve the tool-invocation ability of lightweight models, demonstrating the efficiency of the distillation from advanced models (Qin et al., 2024; Yang et al., 2023b; Liu et al., 2025).

## 2.2 PERSONALIZED LLMS

Personalized LLMs represent LLMs that have been adapted to align with user preferences and characteristics (Zhang et al., 2024c). Existing works mainly focus on the generation of personalized texts or applications in information systems. LLMs are customized as personal conversational AI assistants for various domains, including education (Kasneci et al., 2023; Dan et al., 2023; Park et al., 2024), healthcare (Belyaeva et al., 2023; Abbasian et al., 2024; Jin et al., 2024), finance (Liu et al., 2023; Lakkaraju et al., 2023), legal (Nguyen, 2023), and etc. User profiles are provided via prompts or hidden representation, leading the model to generate personalized text in the dialog. Personalized LLMs have been extensively applied in information systems such as recommender systems (Wu et al., 2023; Chen et al., 2024). LLMs are leveraged as an augmentation module for traditional recommender systems, serving as the content interpreter (Bao et al., 2023; Li et al., 2023; Yang et al., 2023a), the knowledge base (Xi et al., 2024; Wei et al., 2024), or the explainer (Lei et al., 2024; Wang et al., 2023). Also, many works directly deploy LLMs as the direct recommenders via prompt techniques (Lyu et al., 2024; Hou et al., 2024) or fine-tuning (Zhang et al.). Recent work has begun to investigate personalized tool invocation by treating user characteristics as explicit profile features for selecting appropriate tools (Xu et al., 2025; Cheng et al., 2025). However, these approaches overlook a key challenge: in practical settings, users' implicit preferences are not directly observable by the model and instead must be inferred from their behavioral histories.

## 3 PERSONALIZED TOOL INVOCATION

We innovatively consider a practical and high-demand scenario in LLM tool invocation: **personalized tool invocation**. This scenario requires the model to leverage user-specific information when selecting and configuring tools to address user needs. In this chapter, we formally define the task of personalized tool invocation.

Given an LLM with model parameters $\theta$, the general tool invocation task requires the model, when provided with a query $q$ and a set of candidate tools $T$, to select the appropriate tool $t^i$ and populate its corresponding parameters $a_1^i, \cdots, a_m^i$, forming the solution $A = [(t^i, a_1^i, \cdots, a_m^i), \cdots]$.

In conventional formulations of this task, correctness is typically determined by whether the selected tool successfully resolves the query. However, this setting overlooks the fact that multiple tools may solve one problem (e.g., APIs from different platforms with similar capabilities), and that users often have preferences for certain tools–a concept we refer to as **tool personalization**, defined as follows:

**Definition 3.1.** *(Tool Personalization) User $u$ prefers $t^1$ for query $q_1$ and $t^2$ for query $q_2$, where $q_1, q_2$ can be solved by both $t^1$ and $t^2$:*

$$t^1 \succ_{(u,q_1)} t^2; \quad t^2 \succ_{(u,q_2)} t^1 \tag{1}$$

Moreover, in $A$, both tool selection and parameter values are determined solely based on the information contained in the query. For instance, consider the query: "Book me a flight from Los Angeles to New York at 8:45 AM tomorrow". However, in real-world scenarios, users often do not provide such detailed query information. Instead, they may omit certain essential details required for tool invocation, meaning that the model cannot extract all necessary parameters from the query alone. We refer to this personalized scenario as an **Parameter Personalization**, defined as follows:

**Definition 3.2.** *(Parameter Personalization) Given the profile of the user $u$ as $P_u$, the query $q$ and the solution $A$, there exists value $\alpha \in A$, $\alpha \in P_u$ and $\alpha \notin q$. The phenomenon is called parameter personalization, and the query $q$ is called a profile-dependent query.*

Figure 2: Framework of our personalized tool invocation data synthesis: PTool. The pipeline comprises three stages: Tool Generation, User Profile Generation and Query and Answer Generation.

## 4 PERSONALIZED TOOL INVOCATION DATA SYNTHESIS

To address the two challenges in personalized tool invocation mentioned above, we propose an automated data synthesis framework, PTool, for generating high-quality training and evaluation data for personalized tool invocation. The framework consists of three key stages: **Tool Generation**, **User Profile Construction**, and **Query and Solution Generation**, as illustrated in Figure 2. The detailed processes of each stage are described in the subsequent parts of this section.

### 4.1 TOOL GENERATION

To cover the majority of scenarios encountered in daily life, we first constructed a diversified tool library across multiple contexts. Inspired by existing work, we employed an advanced Large Language Model (LLM)-based data synthesis method to generate APIs. Similar to ToolACE, we also developed a structure akin to an API Tree, which allows for the generation of diverse tools.

Specifically, we initially define several demand scenarios from everyday life (e.g., shopping, food delivery, office) as the first-level nodes of the tree. Then, using a depth-first expansion approach, we iteratively refine the functionality at each node until we derive specific API descriptions as the leaf nodes. Notably, in order to generate data that enhances the model's Tool Personalization capability, tools with similar functionalities are required. However, this API Tree expansion approach alone cannot achieve this. Therefore, at the second level of the tree expansion, we introduce the concept of platforms. For each scenario, we generated multiple platforms with distinct characteristics. For example, in the video entertainment scenario, platforms such as YouTube and TikTok were included, where YouTube focuses on long-form videos and TikTok emphasizes short, lifestyle-oriented clips. This enables us to obtain multiple tools with functionally interchangeable capabilities.

### 4.2 USER PROFILE CONSTRUCTION

Personalization requires constructing diverse and realistic user profiles. This process involves three key challenges: (1) defining feature sets relevant to tool invocation, ensuring a structured linkage between user traits and tool selection; (2) maintaining sufficient diversity across profiles to enable generalization to unseen users; and (3) ensuring that profiles contain only observable basic and behavioral information, without incorporating detailed psychological attributes.

**Bottom-up Feature Tree Construction.** To systematically define user profile features, we adopt a tool-driven hierarchical clustering approach. We construct a feature tree, where platform characteristics and tool parameters serve as leaf nodes. Using LLM-based clustering, we recursively merge semantically related parameters, summarizing them into higher-level features until the number of parent nodes at each level falls within a predefined threshold. Notably, we categorize features during initial clustering: explicit basic features (e.g., age, gender) are directly observable, while implicit preferences (e.g., shopping preferences) remain latent and are used in user behavior generation.

**Top-down Characteristic Assignment.** Once the user feature tree is constructed, we encounter the second issue: how to diversify the assignment of values to these features to generate distinct user profiles. When using an advanced LLM to assign $N$ different user features, two options typically arise: one is to assign all features for a single user at a time and repeat this process $N$ times; the other is to assign all features for $N$ users in one pass. The first method incurs higher inference costs and

makes it challenging to avoid repetition across multiple generations, while the second is constrained by the model's context length limitation, especially when N or the number of features is large. Therefore, we adopt a top-down hierarchical assignment based on the tree structure. Specifically, for nodes at the $l$-th layer, we assign $k_l$ different values simultaneously, and for the $(l + 1)$-th layer nodes, the model generates $k_{l+1}$ different values for each parent node's feature value. Thus, for a user feature tree with depth $L$, we can ultimately obtain $N = \prod_{l=0}^{L} k_l$ distinct user profiles. It's important to note that each time the LLM generates $k_l$, this number can be much smaller than $N$, allowing the LLM to generate diverse features in one pass.

**User Behavior Generation.** Once user profiles are assigned, they include both explicit basic features (e.g., occupation, gender, location) and implicit preferences (e.g., price sensitivity, product affinity). However, in real-world scenarios, user preferences are typically inferred through behavioral patterns rather than explicitly stated. To simulate authentic behavioral traits, we employ an LLM-based role-playing approach, where the model generates user actions on various platforms based on their profile and platform characteristics. For instance, given a user's preference for budget-conscious shopping, the model may generate interactions such as "searches for hiking backpacks on Amazon" or "purchases coffee from Walmart for $30." While implicit preferences remain unobservable to the model during task execution, they are embedded in prompts when generating tool invocation solutions, ensuring accurate and contextually appropriate tool selection.

### 4.3 QUERY AND SOLUTION GENERATION

For generating query-solution pairs, we adopt a multi-agent collaborative approach, involving two agents: the user agent and the assistant agent. The user agent generates queries by role-playing based on the user profile, while the assistant agent generates tool invocation solutions. The user agent's role information includes both basic and implicit features, as these provide a more accurate user representation than explicit behavioral features.

Given that a user's platform preferences may vary across queries, we explicitly incorporate platform information into the user agent's prompt. This enables the agent to generate queries aligned with the user's platform preferences. Additionally, we instruct the user agent to avoid revealing profile information in the queries, ensuring the generation of profile-dependent queries as well.

To ensure the correctness of tool invocations, we employ a two-tier verification strategy: rule-based validation and model-based verification. Rule-based validation checks the format of tool invocations to prevent issues such as unresolvable results or hallucinated tools and parameters. Model-based verification inputs the user profile, query, and solution triples into the LLM to verify parameter correctness, detect hallucinations, and assess whether the solution effectively resolves the query. Furthermore, to ensure evaluation accuracy, we manually inspect tool invocation parameters. These parameters are annotated as profile-related or query-related, indicating whether they originate from the user profile or the query, facilitating more precise error feedback during evaluation.

## 5 EXPERIMENTS

### 5.1 EXPERIMENTAL SETTINGS

**Dataset Details**. We leverage GPT-4-turbo to synthesize the personalized tool invocation dataset via our proposed framework. The overall dataset consists of a total of 80 users and 8,197 queries under 5 scenarios, including shopping, takeout, entertainment, work, and travel. Under each scenario, there are 3 platforms and 24 APIs in each platform as tools. We separate the dataset into training and test sets, randomly selecting all queries of 6 users and about 6% queries of another 74 users to form the test set PTBench. The 6 users will not be visible to models in the training process, termed as untrained. To ensure the quality of the test set, we manually verify each sample. Additionally, we construct a dataset comprising 116 samples from two unseen scenarios—finance and lifestyle—which were not exposed to the models during training, to evaluate their generalization capability. The statistics are illustrated in Appendix A.2.

**Evaluation**. We first evaluate the format accuracy by checking if the model's output can give formatted output, verifying instruction following ability. The solution of each sample comprises two parts: platform and tool invocation. The models are required to select the correct user-preferred plat-

Table 1: Comparison with baseline models on PTBench in terms of accuracy. **Bold** and underline represent the best and the 2nd best results. **Tool-P** denotes the tool personalization. *T-\** denotes the correctness \* in tool invocation. **DS-R1-Dis** is the abbreviation of DeepSeek-R1-Distill.

| Type | Model | Format | Tool-P | Param Value | | Tool Invocation | | | Overall | | |
|------|-------|--------|--------|-------|---------|--------|---------|---------|---------|-----------|---------|
| | | | | *Query* | *Profile* | *T-name* | *T-param* | *T-value* | *Trained* | *Untrained* | *Overall* |
| API | GPT-4-turbo | **97.78** | 54.84 | **81.23** | 68.32 | 91.78 | 77.09 | **35.18** | 18.34 | 18.56 | 18.47 |
| | GPT-4o | 90.12 | 44.84 | 71.44 | 61.04 | 82.83 | 69.91 | 28.69 | 13.50 | 17.08 | 15.51 |
| | Deepseek-v3 | 90.95 | 52.80 | 73.09 | 64.16 | 84.60 | 75.30 | 30.85 | 17.08 | 17.57 | 17.36 |
| | Deepseek-r1 | 81.99 | 48.19 | 63.04 | 58.06 | 73.76 | 62.94 | 26.24 | 14.77 | 14.94 | 14.86 |
| | Qwen-max | 76.92 | 49.46 | 60.94 | 54.40 | 70.91 | 58.43 | 23.48 | 14.56 | 17.07 | 15.97 |
| | Claude-3.5-sonnet | 96.86 | 58.26 | 78.24 | 65.04 | 71.10 | 64.45 | 23.26 | 13.29 | 13.95 | 13.67 |
| OSS | DS-R1-Llama-8B | 64.27 | 30.19 | 38.23 | 30.12 | 50.80 | 38.02 | 9.81 | 4.85 | 3.94 | 4.34 |
| | DS-R1-Qwen-7B | 60.95 | 14.69 | 23.41 | 10.39 | 36.56 | 21.13 | 2.21 | 0.42 | 0.66 | 0.55 |
| | Qwen2.5-7B-Inst | 78.58 | 37.95 | 61.32 | 41.65 | 68.33 | 54.30 | 18.37 | 7.17 | 7.55 | 7.38 |
| | Llama-3.1-8B-Inst | 88.65 | 40.53 | 66.48 | 51.41 | 79.97 | 62.52 | 21.33 | 9.29 | 9.85 | 9.60 |
| | Mistral-7B-v0.3 | 85.87 | 39.03 | 55.98 | 37.23 | 66.12 | 35.72 | 14.50 | 6.74 | 5.59 | 6.09 |
| | Hammer2.1-7b | 96.49 | 36.38 | 72.96 | 52.59 | 84.02 | 63.16 | 22.62 | 7.39 | 6.89 | 7.11 |
| | ToolACE-8B | 40.35 | 16.81 | 32.89 | 20.49 | 38.87 | 26.31 | 9.06 | 3.38 | 3.78 | 3.60 |
| | Watt-tool-8B | 37.49 | 22.81 | 27.16 | 19.90 | 34.08 | 22.18 | 8.26 | 5.91 | 4.11 | 4.89 |
| | xLAM-7b-r | 95.29 | 32.85 | 67.94 | 49.68 | 86.88 | 59.34 | 22.17 | 6.96 | 7.71 | 7.38 |
| | Ours | 95.75 | **73.74** | 79.33 | **73.41** | **92.42** | **82.90** | 34.17 | **27.01** | **26.60** | **26.78** |

form and then generate suitable tool invocations. Platform accuracy demonstrates the ability of tool preference understanding. The tool invocation consists of three parts: tool name, parameters, and parameter values, where the parameter values comprise query-related and profile-related parameters. Profile-related parameters require the model to infer from the user profile, evaluating the ability to handle profile-dependent query. We calculate the accuracy of the function name, function parameter, and function value, respectively. The calculations of accuracy are detailed in Appendix A.1.

**Baselines**. We compare the latest open-source models and API-based models, as well as fine-tuned tool-calling models. Open-source models include DeepSeek-R1-Distill-Llama-8B(DeepSeek-AI, 2025), DeepSeek-R1-Distill-Qwen-7B(DeepSeek-AI, 2025), Qwen2.5-7B-Instruct(Team, 2024a;b), Llama-3.1-8B-Instruct (AI@Meta, 2024) and Mistral-7B-Instruct-v0.3(Jiang et al., 2023). API-based models include GPT-4-turbo[1], GPT-4o[1], Deepseek-v3(DeepSeek-AI, 2024), Deepseek-r1(DeepSeek-AI, 2025), Qwen-max(Team, 2024b) and Claude-3.5-sonnet[2]. Models fine-tuned for tool-calling include Hammer2.1-7b(Lin et al., 2024), ToolACE-8B(Liu et al., 2025), watt-tool-8B[3] and xLAM-7b-r(Zhang et al., 2024b; Liu et al., 2024; Zhang et al., 2024a).

**Implementation Details**. To validate the effectiveness of our model, we conducted various experiments by training LLMs with the synthesized dataset. We train the open-source LLM, Qwen2.5-7B-Instruct(Team, 2024a;b), in the supervised fine-tuning (SFT) manner. Due to limited resources, we adopt the parameter-efficient LoRA(Hu et al., 2022) training strategy to fine-tune the model. As for the hyper-parameters setting, we set the rank as 8, alpha as 16 learning rate as $10^{-4}$, LR scheduler as cosine, WarmUp Ratio as 0.1 and epoch as 1 for all modules in the model.

## 5.2 MAIN RESULTS

The overall results are illustrated in Table 1. The detailed results of trained and untrained users are presented in Appendix A.2. We have the following findings according to the results:

*Finding 1:* API-based large models significantly outperform smaller OSS models across various dimensions, including format compliance, tool preference capabilities, and tool invocation abilities. This aligns with the findings of most benchmarks, primarily attributed to the enhanced capabilities enabled by the larger scale of model parameters.

---

[1]https://chatgpt.com

[2]https://www.anthropic.com

[3]https://ollama.com

Table 2: Ablation of user profile on PTBench. The models are trained with various variants. The input in evaluation remains consistent with the training input.

| Data | Untrained | Trained | Overall |
|---|---|---|---|
| All | **26.60** | **27.01** | **26.78** |
| All w/o Basic | 9.69 | 24.26 | 16.06 |
| All w/o History | 24.63 | 25.31 | 24.93 |
| All w/o Basic&History | 5.91 | 7.81 | 6.74 |

*Finding 2:* Most models fall short on the tool-preference task, including the state-of-the-art model–GPT-4-turbo, indicating the high complexity of selecting a suitable one from several similar tools according to the user profile. Our model outperforms nearly all models in all aspects by a considerable improvement, presenting the necessity of personalized tool-invocation enhancement.

*Finding 3:* Our model demonstrates a significant improvement in its performance across various tasks on PTBench. Notably, the enhancement in the Tool Preference task is particularly pronounced when compared to the pre-trained Qwen2.5-7B-Instruct model. This also indicates that, even without additional manual verification of the training data, the model achieves a high accuracy, demonstrating the effectiveness of the proposed synthesis framework. Additionally, our model shows a significant improvement on untrained users, presenting the generalization of the model.

*Finding 4:* All models exhibit lower accuracy on profile-dependent parameter values compared to query-dependent parameters, indicating that inferring parameters from the profile presents a greater challenge. While our trained model does not surpass GPT-4-turbo in accuracy on query-dependent parameters, it outperforms larger models on profile-dependent parameters. Furthermore, the improvement over the pre-trained Qwen2.5-7B-Instruct model is more substantial, demonstrating the effectiveness of our data generation framework in handling the query-dependent query tasks.

## 5.3 ABLATION STUDY

To investigate the importance of various parts in our synthesized user profile, we conduct the ablation study on the user profile, including 4 variants on the user profile:

- **All.** All information in the user profile is used, including basic features and behavioral history.

- **All w/o Basic.** Basic features are omitted.

- **All w/o History.** The behavioral history is given.

- **All w/o Basic&History.** Both basic features and behavioral history are omitted.

First, We use the four dataset variants to train and then evaluate the model with the consistent input. The results are reported in Table 2. From the result, we can observe that the existence of user history and basic features hold contributions to the overall performance of the model to an extent.

Additionally, we conduct experiments under two settings: (1) train the model with the All variant and evaluate the model with the four variants, illustrated in Figure 3a; (2) train the model with the four variants and evaluate the models with the All variant, illustrated in Figure 3b. The results exhibit that the model shows poor performance in the tool preference task when lacking user history information in training or evaluation. On the other hand, the accuracy of tool invocation suffers when basic features are absent, led by the challenging profile-dependent query task.

To further confirm that the curated instructions can only be completed with personalized information, we conducted an additional experiment where all personalized information was removed from the instructions. As shown in Table 3, model performance decreases in all settings compared to main results, with the most pronounced decline observed in the precision of tool values. These results confirm that personalized information is crucial and indispensable for achieving optimal performance.

Table 3: Evaluating Model Performance Without Personalized Information.

| Model | Format | Platform | T-name | T-param | T-value | Overall |
|---|---|---|---|---|---|---|
| GPT-4o | 92.80 | 48.29 | 87.26 | 64.54 | 8.59 | 5.35 |
| Deepseek-v3 | 98.34 | 51.25 | 91.69 | 77.28 | 10.16 | 5.72 |
| Qwen2.5-7B-Instruct | 90.58 | 44.32 | 82.64 | 64.64 | 8.96 | 4.16 |
| Llama-3.1-8B-Instruct | 95.29 | 43.31 | 87.35 | 69.07 | 9.23 | 3.97 |
| Ours | 95.57 | 52.34 | 96.51 | 87.55 | 6.18 | 5.91 |

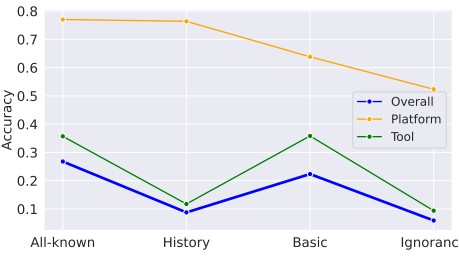 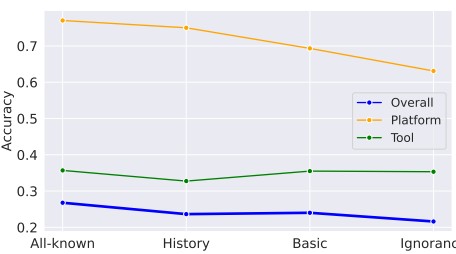

(a) User profile ablation in evaluation.  (b) User profile ablation in training.

Figure 3: Ablation study on user profile in evaluation and training, respectively.

## 5.4 ERROR ANALYSIS

To gain deeper insights into the types of errors made by the models during the evaluation, we conduct investigations into the error types on our model, GPT-4-turbo, and Qwen2.5-7B-Instruct. We only analyze solutions with the correct format.

We analyze the function errors generally and divide them into 6 categories: wrong tools, missing tools, excessive tools, missing parameters, excessive parameters, and wrong parameters. The results are shown in Figure 4. From the pie chart, it is evident that filling the correct parameters is more challenging than the selection of the correct tools. After training with our synthesized data, the model is more familiar with the candidate tools, demonstrating less error percentage in tool selection.

## 5.5 FURTHER ANALYSIS

**Model Scaling.** For the purpose of analyzing the influence of model size on the performance of our trained model, we utilize models with different sizes in the Qwen2.5 series, including 7B, 3B, 1.5B and 0.5B. The results are shown in Figure 5. We can observe that the 1.5B and 0.5B model only show slight improvement from the training, while 3B and 7B model gain substantial improvement from the training. This demonstrate that the personalized tool invocation is a high-level capability of LLMs, requiring a certain scale of parameters.

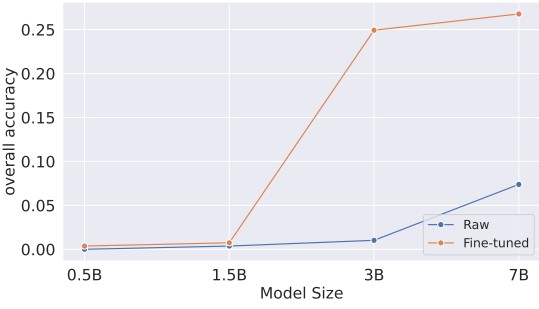 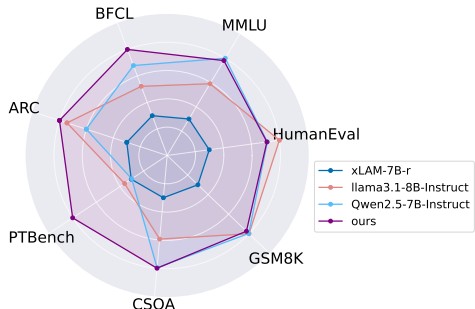

Figure 5: Study of model scaling. (Base model: Qwen2.5-series.)

Figure 6: General Capabilities Analysis. (Base model: Qwen2.5-7B-Instruct.)

Table 4: Models performance results on new scenarios.Bold represents the best result.

| Model | Format | Platform | T-name | T-param | T-value | Overall |
|---|---|---|---|---|---|---|
| GPT-4-turbo | 91.80 | 38.37 | 76.32 | 55.30 | 16.92 | 5.48 |
| Deepseek-v3 | 94.10 | 47.05 | 86.80 | 77.19 | 20.10 | 7.77 |
| Qwen2.5-7B-Instruct | 83.48 | 28.44 | 60.09 | 24.31 | 4.13 | 1.84 |
| Llama-3.1-8B-Instruct | 94.50 | 30.73 | 71.10 | 52.30 | 9.63 | 2.29 |
| Hammer2.1-7b | 94.04 | 33.03 | 75.23 | 37.16 | 9.63 | 4.59 |
| xLAM-7b-r | 98.62 | 26.61 | 83.49 | 41.28 | 10.55 | 3.67 |
| Ours | **100.00** | **73.39** | **88.99** | **75.69** | **26.15** | **18.35** |

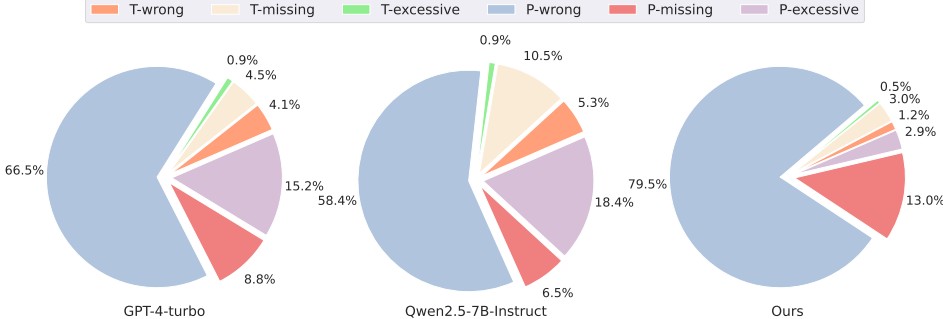

Figure 4: Error Analysis on PTBench. T-wrong, T-missing, and T-excessive represent wrong tools, missing tools and excessive tools, respectively. P-missing, P-excessive and P-error represent missing parameters, excessive parameters and wrong parameters, respectively.

**Generalization to Unseen Scenarios.** To further examine the generalizability of our model beyond the five common scenarios, we conduct additional evaluation on unseen domains. Specifically, we synthesized 218 samples covering three new scenarios: finance(42 samples), lifestyle(74 samples), and knowledge(102 samples). As shown in Table 4, our model consistently outperforms the baselines under all domains, demonstrating strong robustness and adaptability. These results provide evidence that the synthesized framework can be extended to a broader range of real-world scenarios. Further derailed results are shown in Appendix A.2.

**General Capabilities.** In order to validate that our synthesized data does not introduce negative effects on the model's general capabilities, we employ a diverse set of benchmarks to assess the performance from different perspectives, Including general ability (MMLU (Hendrycks et al., 2021a;b)), coding (HumanEval (Chen et al., 2021)), math (GSM8K (Cobbe et al., 2021)), reasoning (CommonSenseQA (Talmor et al., 2019)), abstract reasoning (ARC (Chollet, 2019)), and basic function-calling (tool-invocation) ability (BFCL non-live (Yan et al., 2024)). xLAM-7B-r, LLaMA-3-8B-Instruct, Raw Qwen2.5-7B-Instruct serve as baselines. The results are shown in Figure 6. From the figure, it is evident that there is no significance deterioration on abilities of our model compared to the raw model Qwen2.5-7B-Instruct. Nonetheless, our model gains a notable improvement on BFCL non-live, These findings suggest that our approach effectively enhances personalized functional calling capabilities without compromising the underlying LLM's other abilities.

## 6  CONCLUSION

In this work, we introduce the concept of personalized tool invocation, which encompasses two primary tasks: tool preference and profile-dependent queries. These tasks require the model's ability to understand the user's profile, select preferred tools based on historical behavior, and extract tool parameters from user information. To enhance and evaluate the model's personalized tool invocation capabilities, we propose a data synthesis framework and create a benchmark, PTBench, by manually inspecting a subset of the generated data. Extensive experimental evaluations assess the personalized tool invocation abilities of existing models, confirming the effectiveness of our synthesized data and its harmlessness to other model capabilities.

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

# A EXPERIMENTS

## A.1 EVALUATION METRICS

We categorize the PTBench metrics according to the key competencies required in personalized tool-use scenarios as follows:

- **General Tool Use Capabilities.** This includes assessing format adherence through *Format Accuracy*, selecting correct tool through *Tool Name Accuracy*, specifying valid parameter names through *Tool Parameter Accuracy*, and filling correct values through *Tool Parameter-Value Accuracy*.
- **Personalized Tool Use Capabilities.** This includes (i) tool preference personalization, measured by *Platform Accuracy*, and (ii) profile-dependent query personalization, measured by the *Profile-related Parameter-Value Accuracy*.

The calculation of various metrics in PTBench are formulated as follows:

- **Format Accuracy** indicates the instruction-following ability.

$$format\_acc = \frac{\#parsable\,samples}{\#total} \tag{2}$$

- **Platform Accuracy** indicates the tool preference recognition ability.

$$platform\_acc = \frac{\#correct\,platform\,samples}{\#total} \tag{3}$$

- **Query-related Parameter-Value Accuracy** indicates the ability to extract values from query.

$$query\_param\_acc = \frac{\#correct\,query\,params}{\#total\,query\,params} \tag{4}$$

- **Profile-related Parameter-Value Accuracy** indicates the ability to extract values from profile.

$$profile\_param\_acc = \frac{\#correct\,profile\,params}{\#total\,profile\,params} \tag{5}$$

- **Tool Name Accuracy** indicates the tool selection ability.

$$tool\_name\_acc = \frac{\#correct\,name\,samples}{\#total} \tag{6}$$

- **Tool Parameter Accuracy** indicates the tool comprehension ability.

$$tool\_param\_acc = \frac{\#correct\,param\,samples}{\#total} \tag{7}$$

- **Tool Parameter-Value Accuracy** indicate the value extraction on context ability.

$$tool\_value\_acc = \frac{\#correct\,value\,samples}{\#total} \tag{8}$$

- **Overall Accuracy on Trained Users** indicate the personalized tool ability on trained users.

$$trained\_overall\_acc = \frac{\#correct\,trained\,samples}{\#trained\,total} \tag{9}$$

- **Overall Accuracy on Untrained Users** indicate the personalized tool selection ability on trained users.

$$untrained\_overall\_acc = \frac{\#correct\,untrained\,samples}{\#untrained\,total} \tag{10}$$

- **Overall Accuracy** indicate the overall personalized tool selection ability.

$$overall\_acc = \frac{\#correct\,samples}{\#total} \tag{11}$$

Table 5: Statistics of our synthesized dataset. The samples in the test set are verified by human annotators. Trained and untrained represent the user profiles present and absent in the training set, respectively. Unseen scenario represents additional data used in generalization study.

| Dataset | #Scenario | #Platform | #API | #User | #Query |
|---------|-----------|-----------|------|-------|--------|
| **Train** | 5 | 15 | 360 | 74 | 7,096 |
| **Test(PTBench)** | 8 | 24 | 576 | 95 | 1,301 |
| –Trained | 5 | 15 | 360 | 74 | 474 |
| –Untrained | 5 | 15 | 360 | 6 | 609 |
| –Unseen Scenarios | 3 | 9 | 216 | 15 | 218 |
| **Total** | 8 | 21 | 576 | 95 | 8,397 |

## A.2 DETAILED RESULTS

**Dataset statistics.** The statistics of our training and test sets are illustrated in Table 5.

**Detailed results on trained and untrained user sets.** The detailed results of the trained and untrained user set on PTBench are illustrated in Table 11 and Table 12, respectively.

**Detailed results on three unseen scenarios.** The detailed results of unseen scenarios (lifestyle, finance, and knowledge) are illustrated in Table 6, Table 7 and Table 8, respectively.

Table 6: Models performance results on lifestyle scenario. Bold represents the best result.

| Model | Format | Platform | T-name | T-param | T-value | Overall |
|-------|--------|----------|--------|---------|---------|---------|
| GPT-4-turbo | 98.65 | 41.90 | 94.60 | 70.27 | 20.27 | 4.05 |
| Deepseek-v3 | 100.00 | 47.30 | 97.30 | **83.78** | 21.62 | 6.76 |
| Qwen2.5-7B-Instruct | 81.08 | 20.27 | 72.97 | 21.62 | 4.05 | 1.35 |
| Llama-3.1-8B-Instruct | 97.30 | 39.19 | 93.24 | 64.87 | 10.81 | 2.70 |
| Hammer2.1-7b | 91.89 | 25.67 | 87.84 | 36.49 | 8.10 | 2.70 |
| xLAM-7b-r | 97.30 | 28.38 | 95.95 | 44.59 | 8.11 | 2.70 |
| Ours | **100.00** | **63.51** | **97.30** | 82.43 | **27.03** | **14.86** |

Table 7: Models performance results on finance scenario. Bold represents the best result.

| Model | Format | Platform | T-name | T-param | T-value | Overall |
|-------|--------|----------|--------|---------|---------|---------|
| GPT-4-turbo | 90.48 | 40.48 | 76.19 | 54.76 | 21.43 | 7.14 |
| Deepseek-v3 | 100.00 | 54.76 | 90.48 | 80.95 | **23.81** | 11.90 |
| Qwen2.5-7B-Instruct | 83.33 | 30.95 | 54.76 | 21.43 | 7.14 | 4.76 |
| Llama-3.1-8B-Instruct | 90.48 | 33.33 | 59.52 | 47.62 | 4.76 | 2.38 |
| Hammer2.1-7b | 92.86 | 38.10 | 80.95 | 38.10 | 7.14 | 7.14 |
| xLAM-7b-r | 100.00 | 26.19 | 90.48 | 35.71 | 11.90 | 7.14 |
| Ours | **100.00** | **73.81** | **90.48** | **80.95** | 21.43 | **16.67** |

## A.3 ADDITIONAL ANALYSIS ON RELATED WORKS.

**Comparison with Existing Personalized Tool Benchmarks.** We compare our benchmark with existing personalized tool-use benchmarks, such as ETAPP (Hao et al., 2025) and ToolSpectrum (Cheng et al., 2025). Despite the concurrency, our benchmark maintains several substantive advantages in terms of scale, personalization coverage, and profile design , which is shown in 9:

- **Broader coverage:** We include substantially more domains and tools.

- **Comprehensive personalization definition:** We jointly define personalization in both tool selection and parameter completion.

Table 8: Models performance results on knowledge scenario. Bold represents the best result.

| Model | Format | Platform | T-name | T-param | T-value | Overall |
|---|---|---|---|---|---|---|
| GPT-4-turbo | 87.38 | 34.95 | 63.11 | 44.66 | 12.62 | 5.83 |
| Deepseek-v3 | 87.38 | 43.69 | 77.67 | 70.87 | 17.48 | 6.80 |
| Qwen2.5-7B-Instruct | 85.29 | 33.33 | 52.94 | 27.45 | 2.94 | 0.98 |
| Llama-3.1-8B-Instruct | 94.12 | 23.53 | 59.80 | 45.10 | 10.78 | 1.96 |
| Hammer2.1-7b | 96.08 | 36.27 | 63.73 | 37.25 | 11.76 | 4.90 |
| xLAM-7b-r | 99.02 | 25.49 | 71.57 | 41.18 | 11.76 | 2.94 |
| Ours | **100.00** | **80.39** | **82.35** | **68.63** | **27.45** | **21.57** |

- **More realistic profile design:** Instead of using basic features + implicit preferences (where implicit preferences are directly provided), we structure profiles as basic features + user history. This design is closer to real-world settings, where users' implicit preferences (e.g., price sensitivity) are not explicitly written but must be inferred from behavioral history. This strengthens the practical value of our data synthesis pipeline.

Table 9: Comparison among personalized tool-use benchmarks.

| Benchmarks | #Tools | #Users | #Samples | Tool P. | Param P. | User Traj |
|---|---|---|---|---|---|---|
| ETAPP | 35 | 16 | 800 | ✓ | ✗ | ✗ |
| ToolSpectrum | 42 | 158 | 1000 | ✓ | ✓ | ✗ |
| **Ours** | **360** | **85** | **1301** | ✓ | ✓ | ✓ |

**Generalization on other personalized tool benchmarks.** To further investigate whether our method generalizes beyond our proposed benchmark PTBench, we additionally evaluate the models on two other personalized tool-use benchmarks: ACEBench (Chen et al., 2025) and ToolSpectrum (Cheng et al., 2025). In addition to general-purpose model Qwen2.5-7B-Instruct, we also include PEToolLLM (Xu et al., 2025) as strong personalized tool-use baselines, ensuring a fair and comprehensive comparison. The results are presented in Table 10. From the results, it is clear that our model achieves consistent and robust improvements across all three benchmarks, demonstrating not only its capability in general personalized tool invocation but also its effectiveness in handling diverse tool-use patterns. These observations collectively show that our synthesized data and training strategy enable the model to generalize well across multiple personalized tool-use benchmarks.

Table 10: Performance on various personalized tool benchmarks. Bold represents the best result.

| Model | ACEBench | ToolSpectrum | PTBench |
|---|---|---|---|
| Qwen2.5-7B-Instruct | 0.58 | 0.1759 | 0.0738 |
| PETool-sft | 0.34 | 0.1648 | 0.0433 |
| PETool-sft-dpo | 0.10 | 0.0133 | 0.0130 |
| Ours | **0.66** | **0.1782** | **0.2678** |

## B  HUMAN-IN-THE-LOOP VERIFICATION

In the human verification stage, we adopt a systematic evaluation and refinement protocol consisting of three main steps.

### B.1  DESIGNING EVALUATION CRITERIA

- **Query Reasonableness**: Ensures that queries include all required parameters, align with user profiles, and exclude meaningless characters.

- **Platform Consistency**: Checks whether the platform preference implied in the query is consistent with the answer. If no explicit platform is specified, historical preferences from the user profile are used for verification.

- **Tool Invocation Accuracy**: Verifies that the invoked tool appropriately addresses the query and that its parameters are correctly specified.

### B.2   HUMAN ANNOTATION AND REFINEMENT

A human annotator reviews queries, answers, and tool invocations against the above criteria, making necessary corrections to ensure overall data quality.

A second annotator categorizes tool invocation parameters into two groups:

- **Query-dependent Parameters**: Explicitly provided in the user query.

- **Profile-dependent Parameters**: Not directly mentioned in the query but inferable from the user profile.

This classification enables a fine-grained evaluation of accuracy on different parameters.

## C   EXAMPLES

To enhance the understanding of the proposed personalized tool invocation, we illustrate an example in Figure 7.

Table 11: Comparison with baseline models on trained users in PTBench. **Bold** and underline represent the best and the 2nd best results.

| Type | Model | Format | Preference | Param Value | | Tool Invocation | | | Overall |
|------|-------|--------|------------|-------------|--|-----------------|--|--|---------|
| | | | Platform | Query | Profile | T-name | T-param | T-value | |
| API | GPT-4-turbo | **0.9831** | 0.5569 | **0.7927** | 0.7080 | 0.9325 | 0.7869 | **0.3502** | 0.1834 |
| | GPT-4o | 0.8840 | 0.4157 | 0.6520 | 0.6164 | 0.8143 | 0.6941 | 0.2637 | 0.1350 |
| | Deepseek-v3 | 0.8903 | 0.5043 | 0.6868 | 0.6508 | 0.8376 | 0.7617 | 0.3059 | 0.1708 |
| | Deepseek-r1 | 0.8376 | 0.4958 | 0.6112 | 0.6317 | 0.7637 | 0.6604 | 0.2574 | 0.1477 |
| | Qwen-max | 0.6941 | 0.4430 | 0.5083 | 0.5162 | 0.6393 | 0.5358 | 0.2152 | 0.1456 |
| | Claude-3.5-sonnet | 0.9662 | 0.5822 | 0.7519 | 0.6794 | 0.7152 | 0.6498 | 0.2236 | 0.1329 |
| OSS | DeepSeek-R1-Distill-Llama-8B | 0.6203 | 0.2891 | 0.3495 | 0.3111 | 0.4958 | 0.3925 | 0.1013 | 0.0485 |
| | DeepSeek-R1-Distill-Qwen-7B | 0.6013 | 0.1519 | 0.2148 | 0.0954 | 0.3503 | 0.1941 | 0.0147 | 0.0042 |
| | Qwen2.5-7B-Instruct | 0.7827 | 0.3882 | 0.5900 | 0.4447 | 0.6856 | 0.5612 | 0.1772 | 0.0717 |
| | Llama-3.1-8B-Instruct | 0.8819 | 0.3797 | 0.6384 | 0.5439 | 0.8039 | 0.6498 | 0.2236 | 0.0929 |
| | Mistral-7B-Instruct-v0.3 | 0.8713 | 0.4198 | 0.5522 | 0.4113 | 0.6645 | 0.3734 | 0.1477 | 0.0674 |
| | Hammer2.1-7b | 0.9641 | 0.3650 | 0.7126 | 0.5468 | 0.8439 | 0.6582 | 0.2257 | 0.0739 |
| | ToolACE-8B | 0.4114 | 0.1709 | 0.3147 | 0.2061 | 0.3987 | 0.2721 | 0.0865 | 0.0338 |
| | Watt-tool-8B | 0.3966 | 0.2405 | 0.2708 | 0.2156 | 0.3586 | 0.2510 | 0.0992 | 0.0591 |
| | xLAM-7b-r | 0.9641 | 0.3586 | 0.6732 | 0.5315 | 0.8881 | 0.6329 | 0.2194 | 0.0696 |
| | Ours | 0.9662 | **0.7826** | 0.7791 | **0.7653** | **0.9409** | **0.8628** | 0.3333 | **0.2701** |

Table 12: Comparison with baseline models on untrained users in PTBench. **Bold** and underline represent the best and the 2nd best results.

| Type | Model | Format | Preference | Param Value | | Tool Invocation | | | Overall |
|------|-------|--------|------------|-------------|--|-----------------|--|--|---------|
| | | | Platform | Query | Profile | T-name | T-param | T-value | |
| API | GPT-4-turbo | **0.9737** | 0.5419 | **0.8266** | 0.6637 | 0.9064 | 0.7586 | **0.3531** | 0.1856 |
| | GPT-4o | 0.9146 | 0.4746 | 0.7596 | 0.6057 | 0.8391 | 0.7028 | 0.3054 | 0.1708 |
| | Deepseek-v3 | 0.9245 | 0.5468 | 0.7629 | 0.6343 | 0.8522 | 0.7455 | 0.3104 | 0.1757 |
| | Deepseek-r1 | 0.8062 | 0.4712 | 0.6443 | 0.5403 | 0.7175 | 0.6059 | 0.2660 | 0.1494 |
| | Qwen-max | 0.8276 | 0.5353 | 0.6828 | 0.5658 | 0.7635 | 0.6207 | 0.2496 | 0.1707 |
| | Claude-3.5-sonnet | 0.9704 | 0.5829 | 0.8046 | 0.6275 | 0.7077 | 0.6404 | 0.2397 | 0.1395 |
| OSS | DeepSeek-R1-Distill-Llama-8B | 0.6601 | 0.3120 | 0.4061 | 0.2935 | 0.5173 | 0.3695 | 0.0953 | 0.0394 |
| | DeepSeek-R1-Distill-Qwen-7B | 0.6158 | 0.1429 | 0.2481 | 0.1106 | 0.3777 | 0.2250 | 0.0279 | 0.0066 |
| | Qwen2.5-7B-Instruct | 0.7882 | 0.3727 | 0.6301 | 0.3943 | 0.6815 | 0.5287 | 0.1889 | 0.0755 |
| | Llama-3.1-8B-Instruct | 0.8900 | 0.4253 | 0.6839 | 0.4906 | 0.7964 | 0.6059 | 0.2052 | 0.0985 |
| | Mistral-7B-Instruct-v0.3 | 0.8489 | 0.3678 | 0.5653 | 0.3416 | 0.6584 | 0.3448 | 0.1429 | 0.0559 |
| | Hammer2.1-7b | 0.9655 | 0.3629 | 0.7420 | 0.5094 | 0.8374 | 0.6109 | 0.2266 | 0.0689 |
| | ToolACE-8B | 0.3974 | 0.1659 | 0.3392 | 0.2039 | 0.3810 | 0.2562 | 0.0936 | 0.0378 |
| | Watt-tool-8B | 0.3580 | 0.2184 | 0.2722 | 0.1859 | 0.3268 | 0.2003 | 0.0706 | 0.0411 |
| | xLAM-7b-r | 0.9442 | 0.3054 | 0.6839 | 0.4695 | 0.8538 | 0.5632 | 0.2233 | 0.0771 |
| | Ours | 0.9507 | **0.7028** | 0.8035 | **0.7096** | **0.9112** | **0.8030** | 0.3481 | **0.2660** |

```
[SYSTEM]
You are given a user profile:

{
  "basic_features":{
        "username":"WineTraveler38",
        ...
  }
  "user_history":{
    "shopping":[{
            "platform":"MegaMart",
            "action":"Purchased a selection of premium imported wines"
        }
    ]
  }
}
```

Here is some platforms under the scenario:

```
[{
        "name":"MegaMart",
        "profile":{
            "product range":"A wide-ranging product from various categories."
            ...
        }
    }
    ...
]
```

Here is some APIs under the platforms:

```
[{
        'type': 'function',
        'function': {
            'name': 'registerUser',
            'description': 'Registers a new user in the application.',
            'parameters': {
                'type': 'object',
                'properties': {
                    'username': {
                        'type': 'string',
                        'description': \"User's chosen username.\"
                    },
                    ...
                },
                'required': ['username', 'password', 'email']
            },
            'response': {
                'type': 'object',
                'properties': {
                    'success': {
                        'type': 'boolean',
                        'description': 'Status of registration.'
                    },
                    ...
                }
            }
        }
    },
    ...
]
```

The user will give you a query. Based on the profile, try to solve the query by using the platforms and APIs. The platform you choose should fit the user profile or the needs of the user's query. All the necessary information are provided in the user profile. DO NOT ask the user for further information. You should respond in the format of {platform:[func1(param1_name = param1_value, param2...), func2...]} No other text MUST be included.

```
[USER]
```
Could you please register an account for me using my username, password and email address, and setting my home location to my place of residence? I prefer not to receive any marketing emails.

```
[ASSISTANT]

{
  MegaMart:[
    registerUser(
      username='WineTraveler38', password='strongpassword123!',
      email='jeanlucbordeaux@email.com', preferredLanguage='French',
      marketingConsent=False, homeLocation='Paris, France'
    )
  ]
}
```

Figure 7: Illustration of personalized tool invocation task. The blue part is the expected output.

