# OpenReview forum: "Advancing and Benchmarking Personalized Tool Invocation for LLMs"
_ICLR.cc/2026/Conference — Submitted to ICLR 2026_

### Official Review · Reviewer_i4Sg · 2025-10-22

**Soundness:** 2
**Presentation:** 3
**Contribution:** 3
**Rating:** 2
**Confidence:** 4

**Summary:**

This paper explores the concept of personalized tool invocation, highlighting the importance of personalization in large language models (LLMs) when performing tool-based reasoning and decision-making. The authors propose PTool, a data synthesis framework that constructs user profiles and simulates behavior to generate high-quality personalized tool-invocation data. Using this framework, they further develop PTBench, a benchmark designed to evaluate personalized tool invocation abilities. They fine-tune open-source models on their synthesized dataset and report improvements in tool preference accuracy and parameter inference without sacrificing general capabilities.

**Strengths:**

1. Good direction. Personalization is indeed an important yet underexplored aspect of LLM-based agents.
2. Framework design. The proposed PTool pipeline (tool generation, profile construction, and query-solution synthesis) is clearly structured and systematically implemented.

**Weaknesses:**

1.	Overclaim of novelty. The paper claims to be the first benchmark for evaluating personalized tool invocation. However, related works such as PEToolLLM (2025.2) [1] have already explored personalized tool use. Moreover, some prevalent benchmarks also include personalization aspects, such as ACEBench [3].
2.	Lack of related work discussion and comparison. The paper omits a thorough comparison with existing benchmarks on personalized tool use [1–3]. For a benchmark paper, it is crucial to clearly articulate how this work differs from prior studies in terms of data structure, personalization mechanisms, and dataset construction methods.
3.	Limited generalization analysis. Since the paper involves model fine-tuning, it is necessary to evaluate generalization across other personalized tool use datasets [1–3]. Otherwise, it cannot be determined whether the proposed model works only for the specific personalization forms seen in PTBench.
4. Disorganized metrics. The authors introduce a large number of fine-grained metrics, but it is unclear whether all are necessary. Some metrics appear overlapping—for instance, Param Value and T-value. Besides, the authors should clarify which metrics are related to personalization. Too many metrics without proper categorization and justification may reduce the precision and interpretability of the evaluation.
5. Dataset release. As this paper proposes a benchmark, it is expected that both the training and test datasets be publicly released to facilitate better understanding of the work.

[1] Q. Xu et al, PEToolLLM: Towards Personalized Tool Learning in Large Language Models

[2] Z. Cheng et al, ToolSpectrum : Towards Personalized Tool Utilization for Large Language Models

[3] C. Chen et al, ACEBench: Who Wins the Match Point in Tool Usage?

**Questions:**

In Figure 6, are the models trained solely on PTBench, or do they also incorporate other general-purpose datasets? In prior works such as ToolGen [1], fine-tuning on tool-related data often results in a noticeable degradation in general-domain performance. However, this paper reports almost no such decline. Could the authors clarify what specific characteristics of the proposed dataset or training methodology help prevent this degradation? Moreover, benchmarks that exhibit significant degradation in [1], such as ARC, are also expected to be reported for a more comprehensive comparison.

[1] R. Wang et al, ToolGen: Unified Tool Retrieval and Calling via Generation.

---

> ### Author Response · Authors · 2025-11-22
>
> [W1&W2&W3] Thank you for highlighting these relevant studies. Our benchmark was released early, and several of these works appeared concurrently, which is why they were not included in the initial submission. We will incorporate a detailed comparison in the revised Related Work section.
>
> Despite the concurrency, our benchmark maintains several substantive advantages in terms of scale, personalization coverage, and profile design:
> - **Broader coverage**: We include substantially more domains and tools.
> - **Comprehensive personalization definition**: We jointly define personalization in both tool selection and parameter completion.
> - **More realistic profile design**: Instead of using basic features + implicit preferences (where implicit preferences are directly provided), we structure profiles as basic features + user history. This design is closer to real-world settings, where users’ implicit preferences (e.g., price sensitivity) are not explicitly written but must be inferred from behavioral history. This strengthens the practical value of our data synthesis pipeline.
>
> | Benchmarks      | Tools | Profiles | Samples | Tool Personalization | Parameter Personalization | User Tracjectory |
> |-----------------|------|----------|---------|----------|----------|---------|
> | ETAPP           | 35   | 16       | 800     | Yes | No | No |
> | ToolSpectrum    | 42   | 158      | 1000    | Yes | Yes | No |
> | Ours            | 360  | 85       | 1301    | Yes | Yes | Yes |
>
> We also evaluated our model on these benchmarks. Results show strong **cross-benchmark generalization**:
>
> | Model                | ACEBench | ToolSpectrum | PTBench  |
> |----------------------|----------|--------------|----------|
> | Qwen2.5-7B-Instruct  | 0.58     | 0.1759       | 0.0738   |
> | PETool-sft           | 0.34     | 0.1648       | 0.0433   |
> | PETool-sft-dpo       | 0.10     | 0.0133       | 0.0130   |
> | Ours                 | 0.66     | 0.1782       | 0.2678   |
>
> These results confirm that our model does not overfit to the specific personalization form of PTBench and exhibits better generalization than baselines across three datasets.
>
>
> [W4] We apologize for the limited discussion in the main text due to space constraints. A full description is provided in the Appendix. Our metrics are organized around the two personalization dimensions:
>
> - Tool personalization: Tool-P for "whether the model selects the correct tool".
>
> - Parameter personalization: Param-Value-Query and Param-Value-Profile for "whether the model fills the correct query-dependent and profile dependent parameters".
>
> We will improve categorization and explanations in the revision to enhance interpretability.
>
> [W5] We confirm that all training data, test data, and the full data-synthesis pipeline will be released upon acceptance to facilitate full reproducibility.
>
> [Q1] All models in Figure 6 are fine-tuned only on PTBench synthetic training data via LoRA. No additional general-purpose corpora were used.
>
> [Q2] ToolGen adopts an aggressive training strategy, involving full-parameter fine-tuning, and vocabulary extension oriented toward tool tokens, which is known to substantially distort general-domain capabilities. In contrast, we use lightweight LoRA fine-tuning, which has been repeatedly shown in prior work [1,2] to maintain strong general-domain performance. This explains why PTBench fine-tuning does not cause noticeable degradation across general benchmarks.
>
> > [1] Liu, Weiwen, et al. "Toolace: Winning the points of llm function calling." ICLR 2025.
> >
> > [2] Biderman, Dan, et al. "LoRA Learns Less and Forgets Less." Transactions on Machine Learning Research.
>
> [Q3] Thank you for the suggestion. We have added ARC evaluations. Results show that our model experiences no degradation, consistent with other general-domain metrics. These numbers will be added to Figure 6 in the revised version.
>
> - Qwen2.5-7B-Instruct (Before finetuning)
>
> |    Tasks    |Version|Filter|n-shot| Metric |   |Value |   |Stderr|
> |-------------|------:|------|-----:|--------|---|-----:|---|-----:|
> |arc_challenge|      1|none  |     0|acc     |↑  |0.5273|±  |0.0146|
> |             |       |none  |     0|acc_norm|↑  |0.5520|±  |0.0145|
> |arc_easy     |      1|none  |     0|acc     |↑  |0.8148|±  |0.0080|
> |             |       |none  |     0|acc_norm|↑  |0.8140|±  |0.0080|
>
> - Ours (After finetuning)
>
> |    Tasks    |Version|Filter|n-shot| Metric |   |Value |   |Stderr|
> |-------------|------:|------|-----:|--------|---|-----:|---|-----:|
> |arc_challenge|      1|none  |     0|acc     |↑  |0.5384|±  |0.0146|
> |             |       |none  |     0|acc_norm|↑  |0.5666|±  |0.0145|
> |arc_easy     |      1|none  |     0|acc     |↑  |0.8237|±  |0.0078|
> |             |       |none  |     0|acc_norm|↑  |0.8262|±  |0.0078|

---

> ### Comment · Reviewer_i4Sg · 2025-11-27
>
> Thanks for the authors’ detailed responses. I am not fully convinced by the framing around “concurrent works.” For example, PEToolLLM was released in February, which is considerably earlier, so the current wording does not feel accurate.
>
> That said, I do appreciate this work. Personalization is an important direction, especially for agents, and the lack of related datasets has been a long-standing issue. This paper helps address that gap and introduces several valuable contributions, such as broader coverage and more realistic profiles. However, the overclaim remains a serious concern, as it is unfair to prior works. In fact, had the paper not claimed to be the first, I might have given a positive score from the beginning. Therefore, if the authors can submit a new version to provide a fair and balanced discussion of its contributions relative to existing work, I am willing to reconsider my score.
>
> Regarding the evaluation metrics, I have read the description in the Appendix, but I also consider that they are difficult to understand. Benchmark metrics should be clearer and ideally allow readers to grasp their meaning at a glance. I wonder whether it would be possible to introduce a small set of macro-level metrics—for example, one capturing tool-use accuracy and another capturing personalization in tool use—with the detailed metrics presented as their components. In many scenarios, such macro metrics are sufficient and more helpful.

---

> > ### Author Response · Authors · 2025-11-29
> >
> > Thanks for your sincere reply.
> >
> > Actually, our work has been submitted to OpenReview on 2025 Febrary on this anonymous link https://openreview.net/forum?id=ODZWHnMSCv. Therefore we claim those works you mentioned are cocurrent works. However, we recognize that since those papers are publicly released earlier then our work, we decide to adopt your suggestion, removing the claim of "the first work" in our revised version and adding citations of those works.
> >
> > As for the evaluation metrics, we fully agree your suggestions so that detailed clarifications are added in the appendix in the revised version.

---

### Official Review · Reviewer_pH9E · 2025-10-23

**Soundness:** 3
**Presentation:** 3
**Contribution:** 3
**Rating:** 4
**Confidence:** 3

**Summary:**

The paper introduces personalized tool invocation for LLMs with two sub-tasks: tool personalization and parameter personalization. It proposes PTool, a three-stage data-synthesis pipeline and builds PTBench. Fine-tuning an OSS model improves platform preference selection and profile-dependent parameter filling; ablations show user history aids tool preference while basic attributes aid parameter completion.

**Strengths:**

Personalized tool use is a pivotal challenge for practical LLM deployment. This paper frames the problem via tool personalization and parameter personalization, proposes a data-generation pipeline accordingly, and substantiates its design with targeted ablation studies.

**Weaknesses:**

1. The first benchmark to evaluate personalized tool invocation claim likely overstates. Contemporary works [1] [2] already target personalized tool use.
2. The scale and diversity of the dataset may be insufficient. With 80 users and 3 platforms per scenario, it’s unclear whether user profile coverage is broad enough and whether platform differences are strongly identifiable across scenarios.
3. The benchmark hides profiles at query time but expects profile-based completion. Add adversarial pairs that contrast parameters guessable by commonsense defaults vs. those only recoverable from profiles, and report discrimination performance to ensure genuine profile use.

[1] Evaluating Personalized Tool-Augmented LLMs from the Perspectives of Personalization and Proactivity.

[2]  ToolSpectrum: Towards Personalized Tool Utilization for Large Language Models.

**Questions:**

How does this work compare against other personalized tool benchmarks.

---

> ### Author Response · Authors · 2025-11-22
>
> [W1] Thank you for pointing out these two concurrent studies. Our benchmark was released relatively early, and these works emerged nearly simultaneously, which is why they were not included in our initial Related Work section. We will add them in the revision.
>
> Despite this, our benchmark maintains several advantages.
> - **Broader coverage**: We include substantially more domains and tools.
> - **Comprehensive personalization definition**: We jointly define personalization in both tool selection and parameter completion.
> - **More realistic profile design**: Instead of using basic features + implicit preferences (where implicit preferences are directly provided), we structure profiles as basic features + user history. This design is closer to real-world settings, where users’ implicit preferences (e.g., price sensitivity) are not explicitly written but must be inferred from behavioral history. This strengthens the practical value of our data synthesis pipeline.
>
>
> | Benchmarks      | Tools | Profiles | Samples | Tool Personalization | Parameter Personalization | User Tracjectory |
> |-----------------|------|----------|---------|----------|----------|---------|
> | ETAPP           | 35   | 16       | 800     | Yes | No | No |
> | ToolSpectrum    | 42   | 158      | 1000    | Yes | Yes | No |
> | Ours            | 360  | 85       | 1301    | Yes | Yes | Yes |
>
> [W2] The table above summarizes open-source personalized tool-invocation benchmarks. Our dataset is advantageous in terms of API scale, sample count, and domain coverage, while maintaining a reasonable number of users. Moreover, our data synthesis framework is highly extensible—additional data can be generated at very low cost by adjusting hyperparameters (e.g., number of users, domain descriptions). All code, data, and model checkpoints will be open-sourced to facilitate community adoption.
>
> [W3] We believe there may be a misunderstanding regarding our evaluation setup. During evaluation, all models have full access to user profiles, including both basic features (e.g., phone number, gender, occupation) and user history (e.g., previously invoked tools). Crucially, implicit preferences are not directly exposed; they must be inferred from behavioral traces, making the evaluation setup more consistent with real-world environments.
>
> Regarding the reviewer's suggestion to separate query-dependent versus profile-dependent parameters, we already conduct related analyses:
> - Table 1 reports accuracy separately for query-dependent and profile-dependent parameter values.
> - Table 2 and Figure 3 present ablation studies, showing that: Basic features contribute more to parameter personalization; User history has a stronger influence on tool personalization accuracy.
>
> These results support that models are genuinely leveraging profile information rather than exploiting query heuristics.
>
> [Q] We also compare strong baseline models on ACEBench and ToolSpectrum. Results demonstrate that our model generalizes well across multiple personalized tool-invocation benchmarks, indicating the robustness and transferability of the data generated by our synthesis framework.
>
> | Model                | ACEBench | ToolSpectrum | PTBench  |
> |----------------------|----------|--------------|----------|
> | Qwen2.5-7B-Instruct  | 0.58     | 0.1759       | 0.0738   |
> | PETool-sft           | 0.34     | 0.1648       | 0.0433   |
> | PETool-sft-dpo       | 0.10     | 0.0133       | 0.0130   |
> | Ours                 | 0.66     | 0.1782       | 0.2678   |

---

### Official Review · Reviewer_tuPa · 2025-10-31

**Soundness:** 3
**Presentation:** 3
**Contribution:** 2
**Rating:** 6
**Confidence:** 2

**Summary:**

This paper introduces the concept of Personalized Tool Invocation, addressing two key challenges: (1) Tool Personalization – selecting among functionally similar tools based on user preferences, and (2) Parameter Personalization – inferring missing tool parameters from the user profile. The authors propose an automated data synthesis framework called PTool, consisting of tool generation, user profile construction, and behavior simulation, and build PTBench, the first benchmark with 1,199 annotated samples for this task. Experiments show that fine-tuning open-source models on the synthesized data significantly enhances personalized tool invocation capabilities without harming general performance

**Strengths:**

1. This paper proposes a paradigm for personalized tool invocation, which is an important research topic.

2. A complete pipeline for data synthesis with rule-based + LLM verification and human-in-the-loop curation yields a usable benchmark.

3. Experiments with both API and OSS models show the effectiveness of the proposed PTool method.

**Weaknesses:**

1. All data are synthetic (seeded by GPT-4-turbo) with limited manual verification scale (1,199 test items), leaving open how results transfer to real-wolrd scenarios.

2. While many baselines are reported, there is no head-to-head against other personalization methods or retrieval-augmented profile resolvers, and no BFCL evaluation.

3. The code and experimental data are not open-sourced.

**Questions:**

1. The unseen-scenario study (116 samples) is promising but small. Could you scale to more domains and report per-domain confidence intervals?

---

> ### Author Response · Authors · 2025-11-22
>
> [W1] We acknowledge the gap between synthetic and real-world data. However, due to privacy constraints, there are currently no publicly available tool-invocation datasets that contain fine-grained user profiles. Our data synthesis framework is explicitly designed to fill this gap. Moreover, synthetic data is widely adopted in recent tool-use and agentic behavior research—from training corpora to benchmarks—and has been empirically shown to improve model capabilities[1,2,3]. Notably, several recently released agentic models[4,5] also rely heavily on synthetic personalized data and report substantial performance gains. Our work follows this methodological trend and further demonstrates that synthetic personalization data can effectively enhance generalization.
>
> > [1] Liu, Weiwen, et al. "Toolace: Winning the points of llm function calling." ICLR 2025.
> >
> > [2] Qin, Yujia, et al. "Toolllm: Facilitating large language models to master 16000+ real-world apis." ICLR 2024.
> >
> > [3] Zhang, Jianguo, et al. "xlam: A family of large action models to empower ai agent systems." NAACL 2025.
> >
> > [4] Team, Kimi, et al. "Kimi k2: Open agentic intelligence." arXiv preprint arXiv:2507.20534 (2025).
> >
> > [5] Team, Meituan LongCat, et al. "Longcat-flash technical report." arXiv preprint arXiv:2509.01322 (2025).
>
> [W2] We have surveyed related work and incorporated PETool, an open-source personalized tool-use model, as a baseline. Retrieval-augmented personalization methods involve a two-stage pipeline that depends on (i) the structure of stored personalized information, (ii) retrieval models, (iii) retrieval strategies, and (iv) a downstream tool-invocation model. These systems introduce many additional variables that fall outside the scope of our study, which focuses on downstream tool invocation models rather than the design of retrieval components. As such, including RAG-based pipelines would not provide a controlled or meaningful comparison for our research objective.
>
> | Model                | ACEBench | ToolSpectrum | PTBench  |
> |----------------------|----------|--------------|----------|
> | Qwen2.5-7B-Instruct  | 0.58     | 0.1759       | 0.0738   |
> | PETool-sft           | 0.34     | 0.1648       | 0.0433   |
> | PETool-sft-dpo       | 0.10     | 0.0133       | 0.0130   |
> | Ours                 | 0.66     | 0.1782       | 0.2678   |
>
> [W3] As stated in the paper, we will release the full codebase, dataset, and trained models upon acceptance to support reproducibility and community development.

---

> ### Author Response · Authors · 2025-11-22
>
> [Q1] To address the reviewer’s concern, we have expanded our unseen-domain evaluation by adding an additional new domain (knowledge) and assessing several strong models on it. The results are shown as below. The extended results consistently show that our model maintains strong accuracy and remains the top-performing system across all unseen domains.
> These findings provide stronger evidence that our synthetic personalization data enables robust cross-domain generalization, which is essential for real-world deployment. We set the temperature to 0.01 and top-k to 1, so the answers remain fixed.
>
> - Lifestyle (74 samples): Apps that support daily well-being with fitness tracking, nutrition, sleep, workouts, mental wellness, and social engagement.
>
> | Model                  | format | platform | T-name | T-param | T-value | overall |
> |------------------------|--------|----------|--------|---------|---------|---------|
> | GPT-4-turbo            | 0.9865 | 0.4190   | 0.9460 | 0.7027  | 0.2027  | 0.0405  |
> | Deepseek-v3            | 1.0000 | 0.4730   | 0.9730 | 0.8378  | 0.2162  | 0.0676  |
> | Qwen2.5-7B-Instruct    | 0.8108 | 0.2027   | 0.7297 | 0.2162  | 0.0405  | 0.0135  |
> | Llama-3.1-8B-Instruct  | 0.9730 | 0.3919   | 0.9324 | 0.6487  | 0.1081  | 0.0270  |
> | Hammer2.1-7b           | 0.9189 | 0.2567   | 0.8784 | 0.3649  | 0.0810  | 0.0270  |
> | xLAM-7b-r              | 0.9730 | 0.2838   | 0.9595 | 0.4459  | 0.0811  | 0.0270  |
> | Ours                   | 1.0000 | 0.6351   | 0.9730 | 0.8243  | 0.2703  | 0.1486  |
>
> - Finance (42 samples): Apps that manage money through secure accounts, payments, budgeting, investments, credit services, and financial insights.
>
> | Model                  | format | platform | T-name | T-param | T-value | overall |
> |------------------------|--------|----------|--------|---------|---------|---------|
> | GPT-4-turbo            | 0.9048 | 0.4048   | 0.7619 | 0.5476  | 0.2143  | 0.0714  |
> | Deepseek-v3            | 1.0000 | 0.5476   | 0.9048 | 0.8095  | 0.2381  | 0.1190  |
> | Qwen2.5-7B-Instruct    | 0.8333 | 0.3095   | 0.5476 | 0.2143  | 0.0714  | 0.0476  |
> | Llama-3.1-8B-Instruct  | 0.9048 | 0.3333   | 0.5952 | 0.4762  | 0.0476  | 0.0238  |
> | Hammer2.1-7b           | 0.9286 | 0.3810   | 0.8095 | 0.3810  | 0.0714  | 0.0714  |
> | xLAM-7b-r              | 1.0000 | 0.2619   | 0.9048 | 0.3571  | 0.1190  | 0.0714  |
> | Ours                   | 1.0000 | 0.7381   | 0.9048 | 0.8095  | 0.2143  | 0.1667  |
>
> - Knowledge (102 samples): Apps that enable learning through content access, search, creation, analytics, collaboration, and personalized recommendations.
>
> | model                  | format | platform | T-name | T-param | T-value | overall |
> |------------------------|--------|----------|--------|---------|---------|---------|
> | GPT-4-turbo            | 0.8738 | 0.3495   | 0.6311 | 0.4466  | 0.1262  | 0.0583  |
> | Deepseek-v3            | 0.8738 | 0.4369   | 0.7767 | 0.7087  | 0.1748  | 0.0680  |
> | Qwen2.5-7B-Instruct    | 0.8529 | 0.3333   | 0.5294 | 0.2745  | 0.0294  | 0.0098  |
> | Llama-3.1-8B-Instruct  | 0.9412 | 0.2353   | 0.5980 | 0.4510  | 0.1078  | 0.0196  |
> | Hammer2.1-7b           | 0.9608 | 0.3627   | 0.6373 | 0.3725  | 0.1176  | 0.0490  |
> | xLAM-7b-r              | 0.9902 | 0.2549   | 0.7157 | 0.4118  | 0.1176  | 0.0294  |
> | Ours                   | 1.0000 | 0.8039   | 0.8235 | 0.6863  | 0.2745  | 0.2157  |

---

### Official Review · Reviewer_smLZ · 2025-10-31

**Soundness:** 3
**Presentation:** 3
**Contribution:** 3
**Rating:** 6
**Confidence:** 4

**Summary:**

This paper introduces the novel task of Personalized Tool Invocation, which encompasses two core sub-tasks: Tool Personalization and Parameter Personalization. The authors develop an automated data synthesis framework, PTool, and construct the first benchmark, PTBench, to evaluate Large Language Models' capabilities in personalized tool invocation. Through fine-tuning open-source models on a large-scale synthesized dataset, experiments demonstrate that their approach significantly enhances model performance on personalized tool invocation tasks without compromising general capabilities.

**Strengths:**

1.	Clearly distinguishing the two key challenges of tool preference and parameter inference, thereby filling a gap in the existing tool learning literature regarding personalization.
2.	Proposes a systematic data synthesis framework (PTool) that covers tool generation, user profile construction, behavior simulation, and query-solution generation. The pipeline is clear and extensible.
3.	Constructs the first benchmark for personalized tool invocation (PTBench), comprising high-quality, human-annotated data and a multi-dimensional evaluation, providing a crucial foundation for future research.

**Weaknesses:**

1.While PTBench contains 1,199 test samples, the training data is primarily synthetic and does not incorporate real user behavior data, which might affect the model's generalization in real-world scenarios.
2.	Evaluation metrics rely heavily on automated measures, lacking subjective assessment of the model's reasoning process or user satisfaction, making it difficult to fully reflect the "human-like" quality of personalized invocations.
3.	The description of the user profile construction process is somewhat brief, particularly regarding how "psychological traits" and "behavioral tendencies" are concretely transformed into operational features, which may hinder reproducibility for readers.

**Questions:**

1.	When constructing user profiles, how did you balance the influence of "basic features" versus "behavioral history"? Is there quantitative analysis identifying which features are most critical for tool preference and parameter inference?
2.	Have you considered extending the PTool framework to more diverse or cross-domain scenarios (e.g., healthcare, finance)? What new challenges might personalized tool invocation face in these domains?

---

> ### Author Response · Authors · 2025-11-22
>
> [W1] We fully agree that real user behavior data is highly sensitive and rarely available in the open domain. This is precisely why we adopt an advanced LLM–based data synthesis pipeline to construct personalized tool-invocation data. While synthetic, prior studies[1,2,3] have repeatedly demonstrated the effectiveness of LLM-generated data for enhancing model capabilities. Moreover, our model is also evaluated on benchmarks containing real user queries, such as BFCL, where it shows competitive and robust performance. These results collectively indicate that the synthetic personalization data produced by our pipeline does not hinder— and in fact strengthens—the model’s generalization ability in real-world scenarios.
>
> > [1] Liu, Weiwen, et al. "Toolace: Winning the points of llm function calling." ICLR 2025.
> >
> > [2] Team, Kimi, et al. "Kimi k2: Open agentic intelligence." arXiv preprint arXiv:2507.20534 (2025).
> >
> > [3] Team, Meituan LongCat, et al. "Longcat-flash technical report." arXiv preprint arXiv:2509.01322 (2025).
>
>
> [W2] Existing work on tool-invocation evaluation primarily relies on objective metrics—namely correctness of tool selection and parameter filling, such as BFCL, T-Eval, ACEBench, et al. To better reflect individualized behavior, our synthetic test set embeds detailed user profiles instantiated by advanced LLMs, including latent psychological traits. This design ensures that user preferences in the test set are accurate, diverse, and semantically coherent, enabling evaluation that meaningfully captures personalized reasoning beyond raw correctness.
>
> [W3] The user-profile construction pipeline consists of three major stages:
> **Bottom-up feature tree construction:**
> We analyze tool attributes and parameter semantics to extract personalization-related factors and organize them into a hierarchical feature tree.
> **Top-down feature assignment:**
> To support diverse user personas, we sample and assign values to high-level categories, progressively propagating them downward to generate coherent user-profile combinations.
> **User-behavior simulation:**
> Since real users rarely expose implicit psychological traits directly, we simulate behavioral histories that reflect these latent preferences.
>
> All code related to profile construction, behavior simulation, and dataset generation will be released alongside the dataset and model upon paper acceptance to ensure full reproducibility.

---

> ### Author Response · Authors · 2025-11-22
>
> [Q1]
> We address this question empirically in our ablation studies. Specifically, we conduct three sets of controlled experiments:
> - Controlling training = inference, ablating basic, history, and basic & history.
> - Using full information during training, while ablating each component at inference time.
> - Using full information during inference, while ablating components during training.
>
> Detailed results are provided in Table 2 and Figure 3. The findings show a clear pattern:
> - Basic features contribute more to personalized parameter filling,
> - Behavioral history contributes more to personalized tool selection accuracy.
> This demonstrates that both components are essential but influence different aspects of personalized tool invocation.
>
> [Q2]
> We have already performed zero-shot evaluations on domains absent from the training set (i.e., finance and lifestyle) as shown in **Table 4**, and our model consistently outperforms all baselines in these unseen domains. During the rebuttal period, we further include an additional unseen knowledge scenario, with domain definitions as follows.
>
> - Lifestyle (74 samples): Apps that support daily well-being with fitness tracking, nutrition, sleep, workouts, mental wellness, and social engagement.
>
> | Model                  | format | platform | T-name | T-param | T-value | overall |
> |------------------------|--------|----------|--------|---------|---------|---------|
> | GPT-4-turbo            | 0.9865 | 0.4190   | 0.9460 | 0.7027  | 0.2027  | 0.0405  |
> | Deepseek-v3            | 1.0000 | 0.4730   | 0.9730 | 0.8378  | 0.2162  | 0.0676  |
> | Qwen2.5-7B-Instruct    | 0.8108 | 0.2027   | 0.7297 | 0.2162  | 0.0405  | 0.0135  |
> | Llama-3.1-8B-Instruct  | 0.9730 | 0.3919   | 0.9324 | 0.6487  | 0.1081  | 0.0270  |
> | Hammer2.1-7b           | 0.9189 | 0.2567   | 0.8784 | 0.3649  | 0.0810  | 0.0270  |
> | xLAM-7b-r              | 0.9730 | 0.2838   | 0.9595 | 0.4459  | 0.0811  | 0.0270  |
> | Ours                   | 1.0000 | 0.6351   | 0.9730 | 0.8243  | 0.2703  | 0.1486  |
>
> - Finance (42 samples): Apps that manage money through secure accounts, payments, budgeting, investments, credit services, and financial insights.
>
> | Model                  | format | platform | T-name | T-param | T-value | overall |
> |------------------------|--------|----------|--------|---------|---------|---------|
> | GPT-4-turbo            | 0.9048 | 0.4048   | 0.7619 | 0.5476  | 0.2143  | 0.0714  |
> | Deepseek-v3            | 1.0000 | 0.5476   | 0.9048 | 0.8095  | 0.2381  | 0.1190  |
> | Qwen2.5-7B-Instruct    | 0.8333 | 0.3095   | 0.5476 | 0.2143  | 0.0714  | 0.0476  |
> | Llama-3.1-8B-Instruct  | 0.9048 | 0.3333   | 0.5952 | 0.4762  | 0.0476  | 0.0238  |
> | Hammer2.1-7b           | 0.9286 | 0.3810   | 0.8095 | 0.3810  | 0.0714  | 0.0714  |
> | xLAM-7b-r              | 1.0000 | 0.2619   | 0.9048 | 0.3571  | 0.1190  | 0.0714  |
> | Ours                   | 1.0000 | 0.7381   | 0.9048 | 0.8095  | 0.2143  | 0.1667  |
>
> - Knowledge (102 samples): Apps that enable learning through content access, search, creation, analytics, collaboration, and personalized recommendations.
>
> | model                  | format | platform | T-name | T-param | T-value | overall |
> |------------------------|--------|----------|--------|---------|---------|---------|
> | GPT-4-turbo            | 0.8738 | 0.3495   | 0.6311 | 0.4466  | 0.1262  | 0.0583  |
> | Deepseek-v3            | 0.8738 | 0.4369   | 0.7767 | 0.7087  | 0.1748  | 0.0680  |
> | Qwen2.5-7B-Instruct    | 0.8529 | 0.3333   | 0.5294 | 0.2745  | 0.0294  | 0.0098  |
> | Llama-3.1-8B-Instruct  | 0.9412 | 0.2353   | 0.5980 | 0.4510  | 0.1078  | 0.0196  |
> | Hammer2.1-7b           | 0.9608 | 0.3627   | 0.6373 | 0.3725  | 0.1176  | 0.0490  |
> | xLAM-7b-r              | 0.9902 | 0.2549   | 0.7157 | 0.4118  | 0.1176  | 0.0294  |
> | Ours                   | 1.0000 | 0.8039   | 0.8235 | 0.6863  | 0.2745  | 0.2157  |

---

### Meta-Review · Area_Chair_hkpT · 2026-01-03

**Summary:**

While this paper presents a complete engineering pipeline and a usable synthetic benchmark, it falls short on novelty, fairness of positioning, and transparency, which are critical for a benchmark submission at ICLR. The rebuttal improves execution but does not sufficiently elevate the contribution beyond existing work. The authors are encouraged to substantially reframe the claims, release the benchmark artifacts, and strengthen real-world validation before resubmission, potentially to a more application- or data-mining-oriented venue.

**Reviewer Concerns:**

The rebuttal provides extensive additional experiments and clarifications, but does not resolve several core concerns that motivated rejection. Most critically, the overclaim of novelty remains only partially addressed: although the authors agree to remove the “first benchmark” claim, the incremental contribution over existing personalized tool-use benchmarks (e.g., PEToolLLM, ToolSpectrum) remains limited.

In addition, benchmark transparency remains insufficient, as neither code nor data are released during review, which is particularly problematic for a synthetic-data benchmark. Concerns about real-world generalization persist, as all evidence continues to rely on LLM-generated synthetic data without comparison to RAG-based or real-user personalization pipelines. Finally, while metric definitions are clarified, the evaluation remains difficult to interpret, lacking clear macro-level indicators.

Overall, the rebuttal improves clarity but does not address the principal issues of novelty, transparency, and realism that are decisive for acceptance.

**Reviewer Scores:**

Reviewer i4Sg (original: 2, reject): Unchanged or slightly increase to 4. The reviewer explicitly states that overclaim and metric clarity remain unresolved, despite appreciating the effort.

Reviewer pH9E (original: 4, borderline reject): Unchanged. Novelty concerns and dataset scale/diversity issues persist.

Reviewer tuPa (original: 6, weak accept): Likely unchanged, as transparency and lack of real-world validation remain problematic.

Reviewer smLZ (original: 6, weak accept): Likely unchanged, though their acceptance is explicitly non-committal (“would not mind if rejected”).

The score distribution therefore remains highly polarized, with the strongest objections tied to principled benchmark standards rather than fixable presentation issues.

---

### Decision · Program_Chairs · 2026-01-26

Reject